# Characterization of Phosphorus Species in Human Dentin by Solid-State NMR

**DOI:** 10.3390/molecules25010196

**Published:** 2020-01-03

**Authors:** Yi-Ling Tsai, Meng-Wei Kao, Shing-Jong Huang, Yuan-Ling Lee, Chun-Pin Lin, Jerry Chun Chung Chan

**Affiliations:** 1Department of Chemistry, National Taiwan University, No. 1, Sec. 4, Roosevelt Road, Taipei 10617, Taiwan; 2Instrumentation Center, National Taiwan University, No. 1, Sec. 4, Roosevelt Road, Taipei 10617, Taiwan; 3School of Dentistry, National Taiwan University Hospital, National Taiwan University, No. 7, Chung San South Road, Taipei 10002, Taiwan

**Keywords:** biominerals, biomineralization, apatite, amorphous calcium phosphate, solid-state NMR

## Abstract

The rat has been considered as an appropriate animal model for the study of the mineralization process in humans. In this work, we found that the phosphorus species in human dentin characterized by solid-state NMR spectroscopy consist mainly of orthophosphate and hydrogen phosphate. Some orthophosphates are found in a disordered phase, where the phosphate ions are hydrogen-bonded to structural water, some present a stoichiometric apatite structure, and some a hydroxyl-depleted apatite structure. The results of this study are largely the same as those previously obtained for rat dentin. However, the relative amounts of the various phosphorus species in human and rat dentin are dramatically different. In particular, stoichiometric apatite is more abundant in human dentin than in rat dentin, whereas the converse is true for disordered-phase orthophosphates. Furthermore, spatial proximity among all phosphorus species in human dentin is identical within experimental error, in contrast to what observed for rat dentin. Although it is not clear how these spectroscopic data could relate to the hierarchical structure or the mechanical properties of teeth, our data reveal that the molecular structures of human and rat dentin at different growth stages are not exactly the same.

## 1. Introduction

Bone and teeth are the major calcified tissues in the human body. The human tooth contains three major parts, viz., enamel, dentin, and cementum [1]. The first detectable crystalline species in the enamel are ribbon-like and could be assigned to octacalcium phosphate (OCP) [2]. In mature enamel, the mineral content is more than 98 wt%. Human dentin, containing 70 wt% of inorganic phase, 20 wt% of organic phase, and 10 wt% of water, has a similar composition to that of bone [3,4]. The major inorganic crystalline phase of dentin or bone is commonly referred to as biological apatite, which is nano-sized and poorly crystallized. Biological apatite is structurally similar to hydroxyapatite (HAp) but non-stoichiometric due to ion substitutions involving sodium, magnesium, and carbonate ions [5]. The chemical formula of HAp is Ca_10_(PO_4_)_6_OH_2_. Due to the structural disorder in biological apatite, that precludes the use of diffraction techniques for its structural characterization, the molecular structure of teeth remains poorly known. In particular, the hydroxyl content of biological apatite is of considerable interest because of its close association with ionic vacancies and substitutions in dentin or bone. However, chemical pretreatment to remove interference from water and organic matrix might complicate the quantitative analysis of the hydroxyl content. Furthermore, it is not trivial to distinguish poorly crystallized apatite from highly disordered-phase calcium phosphate. In this regard, solid-state NMR spectroscopy, which is non-invasive and inherently quantitative, has been proven to be a powerful analytical technique for the study of bone and teeth [6,7,8].

In our previous works [9,10,11], we studied a series of dentin samples taken from Wistar rats using solid-state NMR spectroscopy. Three phosphorus mineral phases, viz., HAp, hydroxyl-deficient apatite (HDAp), and amorphous calcium phosphate (ACP), were identified and quantified. In this work, dentin samples obtained from the third molars of human subjects of different ages were characterized by solid-state NMR techniques. We found that the distribution of the three mineral phases was rather different from that observed in rat dentin. The spatial proximity of the various phosphorus phases in human dentin was also qualitatively different from that of rat dentin.

## 2. Results

### 2.1. XRD, ICP-MS, and TGA Data

The dentin samples prepared from human third molars are referred to as R*_x_* or C*_x_*, where C and R stand for the crown and root parts of the teeth, and *x* denotes the subject’s age in year. The powder X-ray diffraction (XRD) patterns of selected human dentin samples are shown in Figure 1. The two diffraction peaks around 26° and 32° are the typical makers of biological apatite. Overall, the crystallinity of the R*x* samples was rather poor, consistent with what we would expect for biological apatite. The exceptional high crystallinity observed for the pattern of C_22_ was attributed to residual enamel in the samples. To analyze the content of the organic matter, thermogravimetric analysis (TGA) measurements were carried out (Appendix A). The initial weight loss was attributed to the loss of water. The substantial weight loss occurring at temperatures from 284 to 464 °C was due to the removal of organic matter. As summarized in Table 1, the weight loss of all the R*_x_* samples was consistent with that reported in the literature (20 wt%) [1]. However, weight loss for the C*_x_* samples varied from 10.4 to 18.5 wt%. Again, we attributed these anomalous results to the presence of residual enamel. In our discussion, we will focus on the data for R*_x_*.

The mole fractions of Ca, Mg, and P of our samples determined by inductively coupled plasma mass spectrometry (ICP-MS) are listed in Table 1. The Ca/P molar ratios of the R*_x_* samples were in the range of 1.41 to 1.49, similar to those obtained for rat dentin [9]. On the other hand, the Mg content of human dentin was comparable to that of juvenile rats (3.3%) but significantly lower than that of mature rats (≥5.8%) [9].

### 2.2. Quantification of Phosphorus Species in R_x_ Samples

Appendix A shows the typical ^1^H and ^31^P MAS spectra of the R*_x_* samples. All spectra of our dentin samples had similar spectral features. The ^1^H spectrum showed a broad resonance at around 5.5 ppm. The sharp peak at 1.48 ppm was assigned to type I collagen [12]. Another sharp peak at 1.08 ppm, which is commonly observed in as-prepared calcium phosphate, could be assigned to mobile water located near the mineral surface [13]. The peak at ~0.2 ppm was readily assigned to the OH signal of HAp [13]. The ^31^P spectrum comprised a single resonance at 3.1 ppm. The relevant basic ^31^P NMR parameters acquired for the R*_x_* samples are summarized in Table 2. To estimate the total phosphorus content per unit mass for R*_x_*, we compared the signal intensity of the ^31^P MAS spectra with that of pure HAp spectrum. As shown in Table 2, the amount of total phosphorus units in R*_x_* was about 0.85 relative to pure HAp, similar to the data obtained for 24-month-old Wistar rats [9].

Although the phosphorus species present in biominerals are complex, the ^31^P signals of those in close proximity of hydrogen could be eliminated by a variety of NMR dipolar dephasing techniques [7]. One simple approach is to exploit the spin-echo technique under the rotary resonance condition [14]. As illustrated for HAp (Appendix A), the residual signal was ~2% of the total ^31^P signal when the dephasing time was set to 10 ms. By contrast, the intensity of the residual signal acquired for R*_x_* was ~9%, and its chemical shift (2.6 ppm) was very close to that of HAp. Thus, we assigned the residual signal to HDAp. To quantify the signal of HDAp, we had to take into account signal attenuation due to the transverse spin–spin relaxation (*T*_2_) effect. Thus, we repeated our measurements for a series of dephasing times of 10, 12, 14, 16, and 18 ms. Figure 2 shows that the attenuation of the ^31^P spin-echo signals as a function of the dephasing time could be described by an exponential decay. The intensity extrapolated at vanishing spin-echo delay, which was presumably free from the attenuation by *T*_2_ relaxation, was used to quantify the signal of HDAp.

In general, the resolution of one-dimensional ^31^P NMR spectroscopy is not good enough to resolve the signals of the various phosphorus species present in dentin or bone. Fortunately, heteronuclear correlation (HETCOR), a two-dimensional method, is well suited to probe phosphorus species in close proximity to hydrogen, such as HAp and ACP [6,15]. In particular, it has been shown that the Lee–Goldburg cross-polarization (LG-CP) technique can be used to achieve polarization transfer with efficient suppression of ^1^H–^1^H spin diffusion [16,17]. A typical HETCOR spectrum acquired for R*_x_* is shown in Figure 3, in which the signals of HAp and the disordered phase were well resolved. To avoid ambiguity, the disordered phase of dentin is referred to as dentin-ACP in our subsequent discussion. The correlation peaks at 0.2 ppm (^1^H) and 3.2 ppm (^31^P) were readily assigned to the HAp component. On the other hand, the dentin-ACP component had a relatively large chemical shift distribution in the ^1^H dimension (5 to 15 ppm), which indicated that dentin-ACP contained both orthophosphate and hydrogen phosphate ions. The intensities of the CP signals can be empirically described by the following equation:
(1)$$M\left(t\right)=M_{0}\left(1-e^{-t/\tau_{\mathrm{CP}}}\right)e^{-t/T_{1\rho }},$$
where *t* is the experimental contact time, *M*_0_ is the intensity factor of arbitrary unit, *τ*_CP_ characterizes the rate constant of the polarization transfer between the ^31^P and the ^1^H species, and *T*_1*ρ*_ reveals the relaxation behavior of the ^1^H species in the spin-locking rf field. Because HAp and dentin-ACP exhibited different dynamics in polarization transfer, we measured a series of HETCOR spectra at different contact times in order to extract the *M*_0_ values of HAp and dentin-ACP (Figure 4). However, the *M*_0_ values were heavily dependent on the experimental conditions such as the spinning frequency and rf spin-locking fields. Nonetheless, the ratio of the *M*_0_ values should provide a good estimate of the relative amounts of HAp and dentin-ACP (Appendix A). The phosphorus species in close proximity of hydrogen was assumed to comprise HAp and dentin-ACP only. Its amount was obtained by subtracting the signal intensity of HDAp from that of total ^31^P. Figure 5 summarizes the amounts of different phosphorus species determined for human dentin; the results of rat dentin are presented for comparison.

### 2.3. Relative Proximity of the Domains of HAp, HDAp, and Dentin-ACP

To investigate the relative proximity of the domains of different phosphorus species, one could exploit homonuclear ^31^P dipolar spin diffusion [11,18]. The relevant pulse sequences and an illustration of the spectrum are shown in Appendix A. The profiles of the spin diffusion experiments are shown in Figure 6. Accordingly, the correlation times characterizing the ^31^P spin diffusion process between any two domains of HAp, HDAp, and dentin-ACP were identical within experimental error. This is in stark contrast to the results for rat dentin, for which the correlation time between HAp and dentin-ACP was found to be significantly larger than the other correlation times [11]. Apparently, the structural model proposed for rat dentin, where the HAp core surrounded by the HDAp shell apatite crystallites is embedded in an amorphous calcium phosphate matrix, is not applicable to human dentin.

### 2.4. Structural Characterization of As-Prepared ACP

In our foregoing sections, the disordered phase of calcium phosphate in dentin are referred to as dentin-ACP. We thought it would be interesting to compare the NMR parameters of dentin-ACP with those determined for the as-prepared ACP. To stabilize the structure of in vitro prepared calcium phosphate [19], Mg^2+^ ions are usually added. By ICP-MS measurements, the Mg mole fraction, defined as Mg/(Mg + Ca + P), was determined to be 13% for our ACP sample, and the (Mg + Ca)/P ratio was 1.65. For any spin clusters, the distance distribution of the spins can be well characterized by the van Vleck second moments (*M*_2_) [20]. In particular, for the same type of spins such as ^31^P, *M*_2_ is given by:
(2)$$M_{2}=\frac{3}{5}{\left(\frac{\mu_{0}}{4\pi }\right)}^{2}\gamma^{4}\hbar^{2}I\left(I+1\right)\sum_{k}\frac{1}{r_{jk}^{6}},$$
where *I* is the spin quantum number of the resonating NMR nuclei, *γ* is the gyromagnetic ratio, *r_jk_* describes the distance of the spin pair under consideration; other symbols carry their usual meanings. The van Vleck second moments are well suited to describe structural information for both crystalline and non-crystalline solids. Experimentally, the *M*_2_ of ^31^P spin systems can be obtained by measuring the ^31^P spin-echo amplitudes as a function of echo time under static conditions [21]. In this study, the results acquired for HAp and the as-prepared ACP are shown in Appendix A. The experimental *M*_2_ values of HAp and ACP were found to be (7.8 ± 0.6) and (3.3 ± 0.5) × 10^6^ rad^2^/s^2^, respectively. Although the molecular structure of ACP is not exactly known, it is presumably formed by the close packing of roughly spherical Ca_9_(PO_4_)_6_ clusters with water molecules filling the interstices [22]. The good agreement between the experimental and the calculated values of HAp (7.7 × 10^6^ rad^2^/s^2^) lent considerable support to the fidelity of the *M*_2_ value obtained for the as-prepared ACP. As indicated by the *M*_2_ data, the average density of the phosphorus atoms in ACP was substantially smaller than that of HAp. However, it should be cautioned that the measured values for other crystalline compounds of protonated phosphate (brushite and archerite) gave a discrepancy of up to 25% (Table 3). This discrepancy was attributed to the insufficient proton decoupling during the spin-echo delay. The excellent agreement in nahpoite appeared to be fortuitous. In other words, because the structure of dentin-ACP contained the species of HPO_4_^2−^, its *M*_2_ value was subject to larger uncertainty. These data for model crystalline compounds are an important reference for our interpretation of the data acquired for dentin-ACP.

### 2.5. Structural Characterization of Dentin-ACP

The spin-echo approach described in the previous section is not applicable to our dentin samples because the ^31^P signals in different proton environments cannot be distinguished under static conditions. As a result, the so-called double-quantum (DQ)-filtered HETCOR was developed for the study of dentin samples (Appendix A) [10]. Empirically, a DQ excitation profile can be analyzed by the equation:
(3)$$I(\tau_{\mathrm{exe}})=A\tau_{\mathrm{exe}}^{2}\exp(-\tau_{\mathrm{exe}}^{2}/B),$$
where *τ*_exe_ denotes the excitation time, *A* and *B* describe the rate of the initial DQ signal buildup and the decay of the DQ signal, respectively. The value of *A* is proportional to the *M*_2_ value of the interacting nuclear spins [23]. Appendix A shows the experimental DQ profile of the as-prepared HAp and ACP. Their extracted parameters of *A* and *B* are summarized in Table 4, together with the results obtained for selected R*_x_* samples. Accordingly, the *M*_2_ values of the HAp component of both R_17_ and R_51_ were comparable to those of pure HAp. On the other hand, the *M*_2_ values of the dentin-ACP component were significantly larger than that determined for the as-prepared ACP. Additional measurements indicated that the values of the dentin-ACP component of human dentin were similar to those determined for dentin of five-month-old rats. While the *M*_2_ value of dentin of 24-month-old rats was larger than that of dentin of 3-week-old rats by 50%, the *M*_2_ values of human dentin were about the same for samples from subjects of different ages. In other words, in human dentin, the density of the phosphorus species in the disordered phase would not vary with age.

## 3. Discussion

As in our previous study on Wistar rat dentins [9], four different phosphate species were identified in human dentin, viz., HO–H…O–PO_3_^3–^, HPO_4_^2–^, apatitic PO_4_^3–^, and hydroxyl-deficient apatitic PO_4_^3–^. The first two species were collectively referred to as dentin-ACP. The last two species constituted the phases of HAp and HDAp, respectively. As illustrated in Figure 5, the relative amounts of the phosphorus species in human dentin present several interesting features. First, the amounts of HAp and dentin-ACP in human dentin (0.30, 0.22) were very different from those in rat dentin (0.08, 0.49). The rat incisor, from which the rat dentin was prepared, can grow indefinitely. Because the amount of dentin-ACP has been associated with teeth growth [9], it is reasonable to have less dentin-ACP in human dentin than in rat dentin. Despite the dramatic difference in the relative amounts of HAp and dentin-ACP, the elastic moduli of human and rat dentin are not significantly different [24]. Second, the amounts of HDAp in human and rat dentin were found to be more or less the same. Third, the amounts of the various phosphorus species in human dentin did not vary significantly with age. Contrary to the rapid dentin deposition in the rat incisor with age, the physiological formation of secondary dentin in human teeth is very slow after teeth eruption into clinical occlusal function [25]. Thus, it is not surprising that the amount of the various phosphorus species in human dentin did not vary significantly with age. Although the strength, toughness, and fatigue resistance of human dentin deteriorates with age [26,27], our data indicate that this phenomenon is not correlated with the relative distribution of inorganic phosphate. Previously, weakening of the mechanical strength of human teeth was ascribed to the filling extent of dentin tubules with carbonated apatite [28]. However, we did not observe any significant variation of HAp or HDAp with age in our dentin samples, which were taken from the root of human third molars.

We reiterate that HDAp presented different levels of hydroxyl deficiency [11]. Additional experiments using rotational-echo double resonance (REDOR) [29] indicated considerably smaller amount of HDAp, and no plateau values were obtained when the dephasing period was increased (data not shown). Such large diminishment in the signal intensity cannot not be rationalized by merely the depletion of the OH group in HAp, because our control experiment showed that the signal of HAp could be suppressed readily with the rotary resonance method. Apparently, the HDAp phase in our study contained a hitherto uncharacterized hydrogen-containing species. While it is legitimate to assert that a substantial amount of stoichiometric HAp is present in bone mineral [30], the quantities of the various phosphorus species estimated in bone or dentin should not be taken as quantitative measures of their absolute amounts.

## 4. Materials and Methods

All dentin samples were prepared from human third molars, which were removed in dental treatments completely unconnected with this study. Each tooth was separated into the crown and the root, and the enamel surface was subsequently removed. The samples were washed with saline and ground into powder using a ball mill (Mixer Mills MM 301). Mg-stabilized ACP was prepared by dissolving 1.22 g of MgCl_2_·6H_2_O and 2.66 g of CaCl_2_ in 100 mL of deionized (DI) water. The phosphate source was prepared by dissolving 7.6 g of Na_3_PO_4_·12H_2_O in 100 mL of DI water. The Mg/Ca solution was then added dropwise into the phosphate solution. The precipitate was then centrifuged and washed with DI water three times. The sample was dried at 60 °C overnight and then stored in a desiccator.

X-ray diffraction (XRD) was performed on a PANalytical X’ Pert PRO diffractometer (Malvern Panalytical, UK), using Cu Kα radiation (average λ = 1.5418Å). Thermo-gravimetric (TG) analyses were carried out on a thermogravimetric analyzer (DuPont951) with a heat rate of 10 K/min, and the weight loss was monitored from room temperature to 800 °C. The elemental analysis of calcium, magnesium, and phosphorus was performed on an inductively coupled plasma mass spectrometer (Perkin-Elmer Elan 6000). All the samples were dissolved in 2% HNO_3_ aqueous solution. The ICP-MS standard solutions of 1000 mg/L of P, Ca, and Mg (Merck) were diluted into ppb level for the calibration measurements.

All NMR experiments were performed at room temperature, and the Larmor frequencies of ^1^H and ^31^P were 300.1 MHz and 121.5 MHz, respectively, on a Bruker DSX300 NMR spectrometer (Bruker, Germany) using commercial 2.5 mm and 4 mm probes. The sample was confined to the middle one-third of the rotor volume using Teflon spacers. The variation of magic-angle spinning (MAS) frequency was limited to ±2 Hz using a commercial pneumatic control unit. The spin rate was set to 10 kHz. Chemical shifts were externally referenced to 85% phosphoric acid and tetramethylsilane for ^31^P and ^1^H, respectively. Other experimental parameters were described previously [9,11].

## Figures and Tables

**Figure 1 molecules-25-00196-f001:**
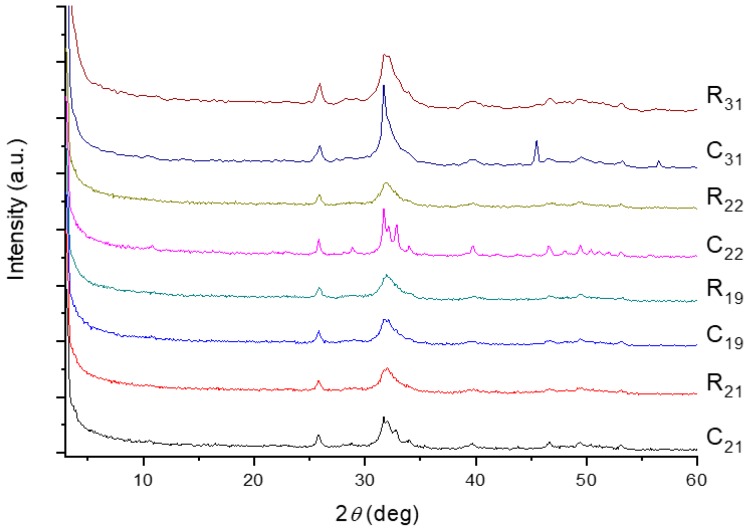
XRD patterns acquired for selected human dentin samples. R_21_ and C_21_ denote dentin extracted from the root and the crown, respectively, of the third molar of a 21-year-old dental patient.

**Figure 2 molecules-25-00196-f002:**
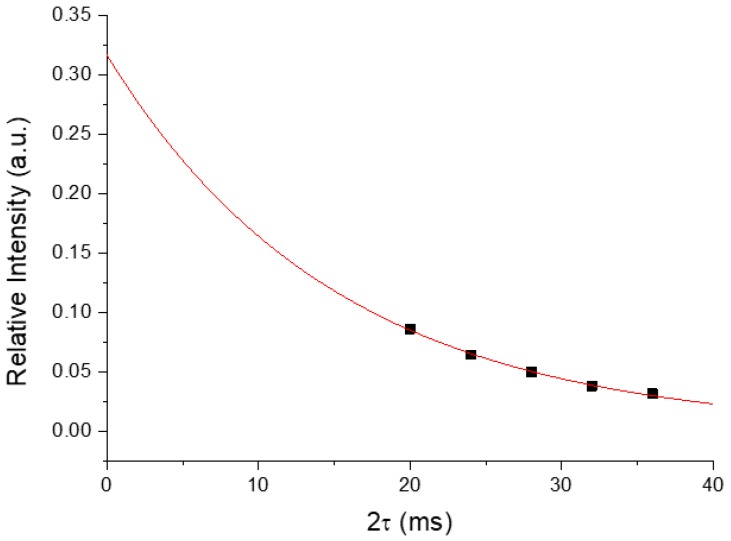
Signal intensities as a function of the dipolar dephasing time acquired for a typical human dentin sample. The intensity extrapolated at 2 *τ* = 0 was used to estimate the amount of hydroxyl-deficient apatite (HDAp). Note that only the data points with *τ* ≥ 10 ms were used for the fitting, because our control experiment showed that the HAp signal was suppressed for *τ* ≥ 10 ms.

**Figure 3 molecules-25-00196-f003:**
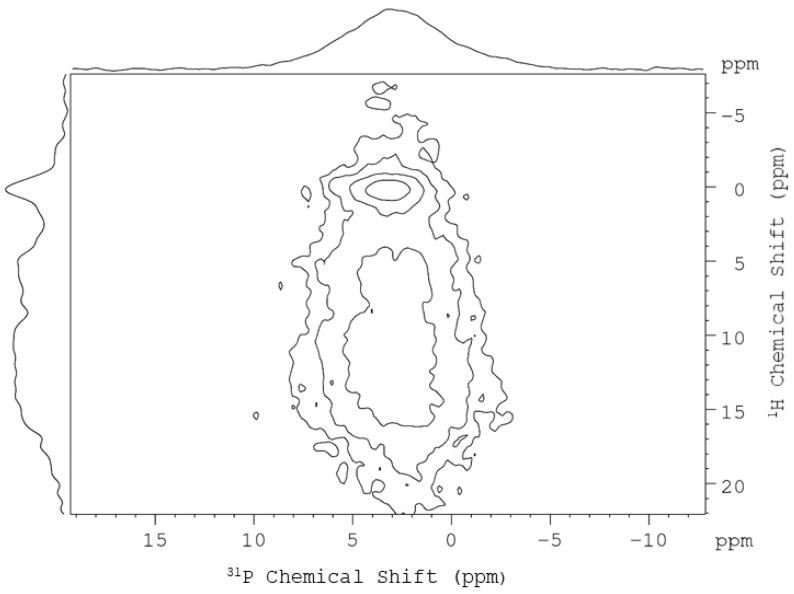
^31^P{^1^H} Lee–Goldburg cross-polarization heteronuclear correlation (LG-CP HETCOR) spectrum of a typical human dentin sample. The contact time was set to 300 μs. The signal integrals of HAp and dentin-ACP were obtained by spectral deconvolution of the projection in the ^1^H dimension.

**Figure 4 molecules-25-00196-f004:**
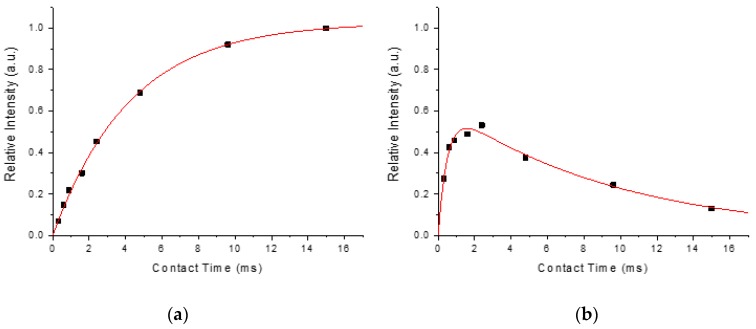
Modulation of the signals of the ^31^P{^1^H} LG-CP HETCOR spectra at variable contact times for the R_21_ sample. The profiles were fitted on the basis of Equation (1), from which the parameters of M_0_, *τ*_CP_, and *T*_1*ρ*_ were extracted. (**a**) HAp; (**b**) dentin-amorphous calcium phosphate (ACP).

**Figure 5 molecules-25-00196-f005:**
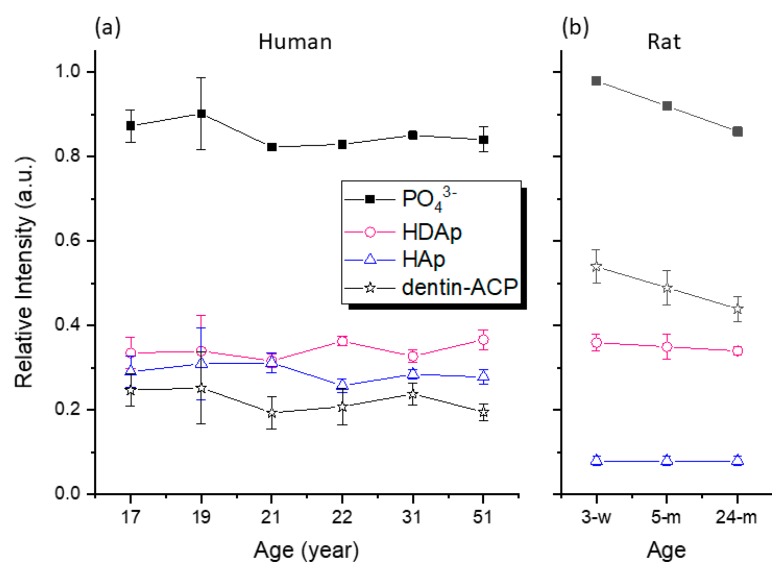
Distribution of the amounts of different phosphorus species normalized on the basis of the sample mineral content. (**a**) Results obtained for human dentin; (**b**) results for rat dentin taken from the literature [8].

**Figure 6 molecules-25-00196-f006:**
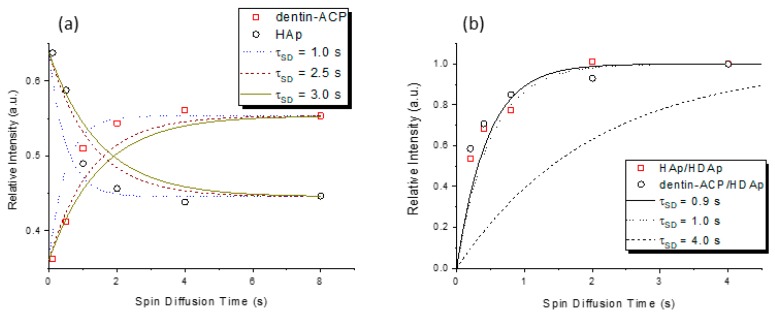
Experimental data of the ^31^P spin diffusion experiments. The simulations were carried out as described in [10] and [17]. (**a**) Results obtained for the transfer between the HAp and the dentin-ACP phases, for which the first CP contact time was set to 12 ms to enhance the initial HAp signal; (**b**) the spin diffusion time constant (*τ*_SD_) extracted for the transfer between HAp and HDAp was similar to that for the transfer between dentin-ACP and HDAp.

**Table 1 molecules-25-00196-t001:** Summary of ICP-MS and TGA data for selected human dentin samples.

Sample	Ca (Mol%)	P (Mol%)	Mg (Mol%)	Organic Matter (Mass%)
C_19_	58.0	39.7	2.3	18.5
R_19_	58.5	39.2	2.3	21.0
C_21_	58.3	39.4	2.3	16.7
R_21_	56.9	38.9	4.2	20.2
C_22_	58.1	40.5	1.5	10.4
R_22_	57.2	40.7	2.1	20.4
C_31_	55.5	41.6	3.0	12.5
R_31_	58.3	39.9	1.9	19.9

**Table 2 molecules-25-00196-t002:** Summary of the NMR parameters of the ^31^P spectra of selected human dentin samples. HAp: hydroxyapatite.

Sample	Chemical Shift (ppm)	Full Width at Half Maximum (ppm)	T_1_ (s)	Signal Integral ^1^ (a.u.)
HAp	2.8	1.5	88	1.0
R_19_	3.1	3.6	116	0.82
R_21_	3.0	2.8	107	0.86
R_22_	3.1	3.0	101	0.83
R_31_	3.1	3.0	126	0.84

^1^ The signal integral was normalized with respect to sample mass and number of scan. The sample mass was corrected for the contribution of organic matter. Other acquisition conditions were identical.

**Table 3 molecules-25-00196-t003:** Summary of the homonuclear van Vleck second moments *M*_2_ of selected crystalline model compounds determined by the spin-echo method under static conditions.

Compound	Chemical Formula	M2 (×10^6^ rad^2^/s^2^)	
		Experimental	Calculated
HAp	Ca_10_(PO_4_)_6_(OH)_2_	7.8	7.7
Brushite	CaHPO_4_·2H_2_O	4.9	6.5
Nahpoite	Na_2_HPO_4_	6.6	6.6
Archerite	KH_2_PO_4_	9.4	7.7

**Table 4 molecules-25-00196-t004:** Summary of the NMR parameters of the ^31^P double-quantum (DQ)-filtered HETCOR spectra acquired for model compounds and selected human dentin samples.

Sample	HAp Component		ACP Component	
	A (ms^−2^)	B (ms^2^)	A (ms^−2^)	B (ms^2^)
HAp	17.1 ± 1.1	5.0 ± 0.3	--	--
ACP	--	--	8.6 ± 0.4	5.2 ± 0.2
R_17_	18.5 ± 1.1	4.6 ± 0.2	13.4 ± 0.7	5.8 ± 0.2
R_51_	18.6 ± 1.2	4.6 ± 0.2	13.1 ± 0.7	5.9 ± 0.2

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
