# Peer review of "Characterization of Phosphorus Species in Human Dentin by Solid-State NMR"

_molecules, 2020, doi:10.3390/molecules25010196_

Round 1

Reviewer 1 Report

The authors of Characterization of Phosphorus Species in Human Dentin by Solid-State NMR describe a thorough investigation of human dentin samples of different ages using 31P solid-state NMR spectroscopy. This work is a valuable extension to authors’ previous studies on rat dentins.  The ssNMR NMR experiments were well designed to obtain quantitative information about relative amounts of Hap, DHAp, and ACP in the samples.  The results were compared with those previously obtained for the rat dentins. The results are further analyzed and summarized with sufficient cautions. The manuscript is carefully prepared and is very well written. I recommend to publish this paper on molecules with a few minor suggestion as the following:

Minor suggestions:

About the title: since only 31P NMR data were reported on the human dentin samples, a more appropriate title would read as: Characterization of Phosphorus Species in Human Dentin by 31P Solid-State NMR.  

The amount of HDAp was obtained by extrapolating the spin-echo curve to smallest delay time tau (Figure 2). Why this curve is only defined by the measured points greater than 10 ms while the dramatic change in the signal intensity in the range of shorter delay (< 10ms) were not sampled at all? Authors should explain.

Author Response

About the title: since only 31P NMR data were reported on the human dentin samples, a more appropriate title would read as: Characterization of Phosphorus Species in Human Dentin by 31P Solid-State NMR.

Responses: Our NMR data include the 31P{1H} HETCOR and some data are 1H directly detected. Therefore, we intend to keep our original title intact.

The amount of HDAp was obtained by extrapolating the spin-echo curve to smallest delay time tau (Figure 2). Why this curve is only defined by the measured points greater than 10 ms while the dramatic change in the signal intensity in the range of shorter delay (< 10ms) were not sampled at all? Authors should explain.

Responses: We showed in our control experiment that the signal intensity of HAp will be greatly diminished by a dephasing time of 10 ms or longer. Therefore, to quantify the signal of HDAp, we only include the data points greater than 10 ms. This point has been reiterated in the figure caption of the revised manuscript.

Reviewer 2 Report

In this manuscript, Tsai and co-authors used solid-state NMR to characterize the phosphorus species in human detin. This group developed ssNMR methods and applied them to rat detins before. In this work, the authors applied similar ssNMR methods on human detins and compared the results with the previous study on rat detins. They found four phosphorus species in human dentin which were the same as rats. However, the amount of the HAp and dentin-ACP were significantly different in human and rats and the amount of various phosphorus species in human dentin didn’t change as age increased. Overall, this is a very interesting and important study. The results are solid, the discussion is thorough, and the conclusion is convincing. A few suggestions for revision are given below.

The quantification of HAP and dentin-ACP is a little confusing. Based on the curves shown in Figure 4, the highest intensity of dentin-ACP is ~20% comparing with HAp. However, in Figure 5a, the amount of dentin-ACP is ~80% comparing with HAp. The results will be clearer if the authors can report the fitting results, especially the ratio of M0, for samples at different ages in a table.

It is not clear that the signal intensity used to quantify the phosphorus content in table 2 is integral intensity or peak height. I guess the authors probably used the integral intensity since the FWHM is not the same. Please specify that.     

In the 31P spin-diffusion experiments, the authors claimed that 12 ms was used as the contact time of LG-CP to suppress the signal from dentin-ACP. Based on figure S2, the resonance peaks from different species are heavily overlapped in the 1H 1D spectrum. It could be helpful if the authors can attach a spectrum in the SI to show the efficiency of the suppression.

The y-axes of plots in Figure 6 are missing, please fix them.

I am curious that why 1H decoupling is not applied during the 31P acquisition? It looks like most of the phosphorus species are close to 1H since the signal from LG-CP is massive. So, the 1H decoupling may be able to provide a better 31P linewidth.

Although some of the NMR parameters have been reported in previous publications, it will be helpful if the authors can report more details in the caption of figures containing pulse sequences in SI, i.e. the duration and amplitude of each pulse, delay or mixing times, the acquisition times, recycle delays, etc.

Author Response

In this manuscript, Tsai and co-authors used solid-state NMR to characterize the phosphorus species in human detin. This group developed ssNMR methods and applied them to rat detins before. In this work, the authors applied similar ssNMR methods on human detins and compared the results with the previous study on rat detins. They found four phosphorus species in human dentin which were the same as rats. However, the amount of the HAp and dentin-ACP were significantly different in human and rats and the amount of various phosphorus species in human dentin didn’t change as age increased. Overall, this is a very interesting and important study. The results are solid, the discussion is thorough, and the conclusion is convincing. A few suggestions for revision are given below.

The quantification of HAP and dentin-ACP is a little confusing. Based on the curves shown in Figure 4, the highest intensity of dentin-ACP is ~20% comparing with HAp. However, in Figure 5a, the amount of dentin-ACP is ~80% comparing with HAp. The results will be clearer if the authors can report the fitting results, especially the ratio of M0, for samples at different ages in a table.

Responses: We apologize that the data shown in the original Figure 4 was not taken for human dentin. We have carefully checked all the data and the revised Figure 4 show the data acquired for the R21 sample. The raw data of the fitting have been provided in Table S1 in the Supporting Information.

It is not clear that the signal intensity used to quantify the phosphorus content in table 2 is integral intensity or peak height. I guess the authors probably used the integral intensity since the FWHM is not the same. Please specify that.     

Responses: Revised accordingly.

In the 31P spin-diffusion experiments, the authors claimed that 12 ms was used as the contact time of LG-CP to suppress the signal from dentin-ACP. Based on figure S2, the resonance peaks from different species are heavily overlapped in the 1H 1D spectrum. It could be helpful if the authors can attach a spectrum in the SI to show the efficiency of the suppression.

Responses: The required figure has been added as Figure S5 in the revised Supporting Information.

The y-axes of plots in Figure 6 are missing, please fix them.

Responses: Revised accordingly.

I am curious that why 1H decoupling is not applied during the 31P acquisition? It looks like most of the phosphorus species are close to 1H since the signal from LG-CP is massive. So, the 1H decoupling may be able to provide a better 31P linewidth.

Responses: In this study, the spectral resolution in the 31P dimension does not matter because we would observe one single peak only even if 1H decoupling is used. Thus, 1H decoupling was avoided to reduce the duty cycle for our experiments.

Although some of the NMR parameters have been reported in previous publications, it will be helpful if the authors can report more details in the caption of figures containing pulse sequences in SI, i.e. the duration and amplitude of each pulse, delay or mixing times, the acquisition times, recycle delays, etc.

Responses: Revised accordingly.